# Material Utilization of Cotton Post-Harvest Line Residues in Polymeric Composites

**DOI:** 10.3390/polym11071106

**Published:** 2019-06-29

**Authors:** Miroslav Müller, Petr Valášek, Viktor Kolář, Vladimír Šleger, Gürkan Alp Kagan Gürdil, Monika Hromasová, Sergej Hloch, Jaromír Moravec, Martin Pexa

**Affiliations:** 1Department of Material Science and Manufacturing Technology, Faculty of Engineering, Czech University of Life Sciences Prague, Kamycka 129, 165 00 Prague 6-Suchdol, Czech Republic; 2Department of Mechanical Engineering, Faculty of Engineering, Czech University of Life Sciences Prague, Kamycka 129, 165 00 Prague 6-Suchdol, Czech Republic; 3Department of Agricultural Machines and Technologies Engineering, Faculty of Agriculture, Ondokuz Mayis University, Körfez Mah.Atakum 55139, Samsun, Turkey; 4Department of Electrical Engineering and Automation, Faculty of Engineering, Czech University of Life Sciences Prague, Kamycka 129, 165 00 Prague 6-Suchdol, Czech Republic; 5Faculty of Manufacturing Technologies, Technical University of Kosice with a seat in Prešov, Bayerova 1, 080 01 Prešov, Slovakia; 6Faculty of Mechanical Engineering, Technical University of Liberec, Studenstká 2, 461 17 Liberec 1, Czech Republic; 7Department for Quality and Dependability of Machines, Faculty of Engineering, Czech University of Life Sciences Prague, Kamycka 129, 165 00 Prague 6-Suchdol, Czech Republic

**Keywords:** adhesive bonds, biofiller, chemical treatment, cyclic loading, epoxy resin, mechanical properties, SEM

## Abstract

This paper deals with a research focused on utilization of microparticle and short-fiber filler based on cotton post-harvest line residues in an area of polymeric composites. Two different fractions of the biological filler (FCR—reinforced cotton filler) of 20 and 100 µm and the filler with short fibers of a length of 700 µm were used in the research. The aim of the research was to evaluate mechanical characteristics of composites and adhesive bonds for the purpose of gaining new pieces of knowledge which will be applicable in the area of material engineering and assessing application possibilities of residues coming into being from agricultural products processing. Mechanical properties of the composite material produced by a vacuum infusion and tested at temperatures 20, 40, and 60 °C and adhesive bonds which were exposed to a low-cyclic loading, i.e., 1000 cycles at 30% to 70% from reference value of the maximum strength, were evaluated. Composite systems with the FCR adjusted in 5% water solution of NaOH showed higher strength values on average compared to untreated FCR. Unsuitable size of the FCR led to a deterioration of the strength. The filler in the form of 700 FCR microfibers showed itself in a positive way to composite materials, and the particle in the form of 20 FCR did the same to adhesive bonds. Results of adhesive bond cyclic tests at higher stress values (70%) demonstrated viscoelastic behavior of the adhesive layer.

## 1. Introduction

Renewable resources are suitable for a substitution of synthetic materials/fillers [1,2]. The utilization of the biological-based filler is a trend in an area of material engineering. Waste production, to which secondary products can also be classified, is a global topic—a number of secondary materials is burned or otherwise liquidated without the possibility of alternative handling. The utilization of all emerging materials during agricultural crops processing can increase the economic efficiency of the whole process and decrease the negative impact to the environment [3,4,5,6]. This fact is presented by many prestigious workplaces which deal with the applicability of natural commodities in material engineering at the present [5,7,8,9,10,11,12,13,14,15,16]. Microparticles of hard organic materials (bamboo powder, powders from fruit stones, fillers from post-harvest line in the Czech Republic, etc.) and vegetable fibers (fibers of sisal, banana tree, jute, flax, hemp) are used which can substitute synthetic materials in many cases [1,2,17,18,19,20,21,22,23]. The substitution of synthetic fillers with natural-based materials has occurred in the area of the composite materials recently [1,2,17,18,19,20,21,22,23].

Natural resources benefit greatly from the potential of biological structures, which have been imitated in a number of synthetic material applications; just the structure of natural/vegetable fibers is an example [13]. Natural reinforcement utilization leads to a decrease of the mass and the costs of single components and to an optimization of mechanical properties [3,4,5]. 

The meaningful utilization of secondary materials/waste, such as material utilization, is an economic and environmental alternative to other handling possibilities or a disposal [8,14]. It is necessary to evaluate arising materials in a complex way in terms of applicability. However, many important parameters have not been described in the area of the material engineering at the present. Just experimental programs extend the competitiveness of biological materials with respect to environmental as well as economic aspects. Composites with a biological material reinforcement are the subject of research in many branches, and they have achieved huge development in recent years [9,10,24,25,26,27]. 

Composites containing biological reinforcement belong among a prospective group of composite materials which define the direction of future development in many aspects [11,13,14,16,28]. 

An essential change of mechanical properties can be reached by adding an optimum volume of the filler into the matrix [29,30,31]. Environmental protection and waste recycling have become one of the main topics of the scientific and industrial community in recent years [32]. They have gradually substituted conventional materials [3]. A low mass, easy production, good physical–chemical properties or corrosion resistance, etc. are the advantages of composite materials [33,34]. They are used in different production areas thanks to these properties, namely in the automotive and air industry, in the production of sport equipment, in seafaring and transport, etc. above all [35]. 

Cotton has been the most extended textile fiber in the world at the present and it finds its highest application in the textile industry [7]. However, the cotton plant can be also used in the application called phytoremediation when pollutants are removed from the environment, e.g., at soil polluted by hard metals such as a lead, etc. [36]. 

The waste, i.e., post-harvest residues, comes into being from cotton production. It consists of cotton stems and cotton plant roots above all. Lignocellulose is the main component of these stems. These residues are burned in most cases, but they can find their utilization as a recyclate [37]. However, post-harvest residues show potential in the energy industry as a source of energy. They can be used, e.g., for the production of bioethanol [38] and methane [39] or as activated carbon [40]. Briquette production is another interesting energy utilization of cotton plant post-harvest residues [41,42]. Much of the waste create cotton textile fibers from clothing. Ten millions tons of this waste come into being each year in Europe and America, and there is a great effort to increase the recycling rate and utilization in other applications, e.g., in composites [43,44]. 

Chemical treatment is required in biological materials, which leads to improvement of mechanical properties and, namely, adhesive strength in the composite system. The chemical treatment of coconut fibers by alkali solution is an example [45,46]. 

Composites with cellulose-based filler, such as just cotton, have been widely applied. They are used thanks to their low production costs above all. The improvement of the mechanical properties above all, e.g., tensile strength, flexural strength or material fatigue, is the purpose of these fillers’ utilization [3,43,47,48]. An interesting use of cellulose fibers is, e.g., at cement production, which is reinforced with these fibers for a purpose of the improvement of its mechanical properties [3]. E.g., fibers of bamboo, sisal, jute, coconut, jatropha, etc. are used [26]. Further, composites with a geopolymer matrix, so called geocomposites, are applied. An aluminosilicate inorganic polymer created by polymerization of aluminosilicate by alkali solutions is called geopolymer [26,27,49]. Cotton fibers were used as the filler into the geopolymer composite in a weight concentration from 0.3% to 1% in one research to find if its mechanical properties were improved [26]. Additionally, dust was used for research on composites. Waste fibers from cotton and wool were frozen in liquid nitrogen and milled at different time intervals [50]. The size of dust particles depends on the milling time, and their size ranged around 60 μm [50]. The size of some particles even ranged around 20 μm [50]. The filler into composites in the form of jatropha dust was used in research on abrasive wear of these materials and a two-component epoxy resin was used as a matrix [10,51]. 

Adhesive bonding technology, i.e., the creation of structural adhesive bonds, represents a significant type of technology of diverse materials bonding, e.g., in the automotive industry, in the production of sport equipment or in the agricultural production, at present. 

It follows from the results of various authors that the mechanical properties of particle composites and adhesive bonds with the hybrid adhesive layer (the adhesive containing the filler, namely, biological) depend on a suitable choice of filler, on an interphase between the matrix and the filler, and on the size of used particles and their concentration. All essential factors have to be taken into account at adhesive bonds production [52,53]. 

The cyclic loading essentially influences the service life of the adhesive bonds [54,55]. This loading, viz. material cyclic fatigue, is the most destructive form of mechanical loading in the result. It is an irreversible process which also comes into being at relatively small loading forces [54]. The cyclic loading of the adhesive bonds represents the most frequent cause of degradation of these materials in practice [55]. The adhesive bond strength is influenced not only by the used filler, but also by a transfer of the stress between single particles and the matrix, i.e., if the particles are well wetted with the matrix, then the stress between the matrix and the particles is transferred efficiently, which significantly increases the adhesive bond strength [25,52,53].

The aim of the research was to evaluate the mechanical characteristics of composites and adhesive bonds for the purpose of gaining new pieces of knowledge which will be applicable in the material engineering area, and to assess the application possibility of residues coming into being from agricultural products processing. The mechanical properties of a composite material produced by vacuum infusion and adhesive bonds were evaluated regarding interfacial interaction through SEM analysis. 

The assessment of renewable resources on the basis of cotton post-harvest line residues was performed within the research, i.e., with a focus on waste biomass processing (secondary commodities in a form of fibers, fruit residues, pulp, etc. coming into being during agricultural products processing), namely in terms of its applicability in the area of composite systems.

## 2. Materials and Methods 

### 2.1. Reinforced Cotton Filler (FCR)

The cotton post-harvest line residues come from machine harvesting in the southeast and west part of Turkey, in the Aydin area. However, machine harvesting is cheaper but of lower quality regarding basic product production, i.e., cotton. Used residues containing not only the plant but also the rest of cotton pod are visible in Figure 1A, from which a predominant portion of stems is evident. Cotton plant leaves were removed before harvesting. Figure 1B,C presents the surface of a predominant part of the plant, i.e., the stem. 

A by-product of the cotton post-harvest line was dried for 24 h at a temperature of 105 ± 5 °C. Subsequently, this input material was multistage-grinded in an industrial grinder. This grinded material was sorted by means of a sieve analysis, i.e., a fractionation was performed in a device Haver EML digital plus. These fractions were used as the reinforced cotton filler (FCR) in composite materials. 

Two different fractions of biological FCR on the basis of cotton post-harvest line microparticles 20 µm (Figure 2A) and 100 µm (Figure 2B) and FCR with short fibers of a length of 700 µm (Figure 2C) were used within the research. 

It is obvious from the results of the length/diameter ratio of tested FCR 700 that they are short-fiber composite systems, i.e., the ratio between the length and the diameter of fibers is smaller than 100. The length/diameter ratio of FCR 700 fibers was 6.09—thus, considerably smaller than 100. It is an anisotropic filler having a similar effect as the particle filler (FCR 20 and 100) thanks to this ratio [56]. The tested composite materials belong to a group with discontinuous reinforcement.

Short-fiber and particle composites are used in applications where it is not possible to exactly define in advance the stress acting or this stress is probably the same in all directions [13]. Further, short-fiber composites are applied in situations where there are requirements for easy workability and good mechanical properties [57,58]. 

The particles size was determined by means of an optical analysis with use of Gwyddion program and images from electron microscopy (SEM). The results of this analysis are stated in Table 1. The results of FCR measuring are summarized in histograms—Figure 3, Figure 4 and Figure 5. The results presented in Table 1 and in Figure 3, Figure 4 and Figure 5 were measured. The measured numbers indicate the length (or the longer axis) of the particles. Tests with FCR without treatment and with chemical treatment in 5% water solution of NaOH for 12 h were performed within the research.

### 2.2. Resin—Matrix (MHC)

The structural epoxy resin LH 288 with a hardener H 282 of the company Havel Composites (MHC) was the matrix [59,60]. The ratio of resin LH 288 to the hardener H 282 was 100:23. This resin is a two-component epoxy resin with a low viscosity suitable for laminating technologies, including vacuum infusion. The hardener H 282 is liquid low viscosity hardener for epoxy resins. The hardener is based on cyclic diamine. The properties of the resin and the hardener according to the producer are stated in Table 2. The resin enables good wetting and saturation with the filler.

### 2.3. Composite Material

A preparation of composite boards was performed by vacuum-assisted resin transfer molding (VARTM). Removal of air before the resin enters is the most important operation of vacuum infusion. Vacuum infusion is an action when the vacuum is used for distribution of the resin through layers of FCR. FCR is laid dry into hollow molds of dimensions 200 × 300 × 4 mm (Figure 6A). Other auxiliary materials covered with the vacuum foil are put on FCR. The resin is sucked in by means of a tubing system and gelcoats. Gelcoats are modified resins which are applied in the liquid state on molds and are used to secure high-quality surface treatments of the composite material visible surface. The vacuum infusion belongs among technologies suitable for composite materials production. The production process through vacuum infusion consists of following technological operations: preparation of the molds, preparation of FCR, gelcoat, putting auxiliary materials into the molds, vacuum foil installation, vacuum check, and production itself. The composite board was prepared with 40 wt.% of FCR. FCR without chemical treatment and with alkali treatment in 5% solution of NaOH for 12 h was used for the research. 

A vacuum pump was used for the vacuum process. It drains air from closed molds and creates a vacuum (Figure 6B). The vacuum 0.8 to 1 bar was developed for the vacuum infusion. Minimum porosity and good distribution of the matrix are advantages of this technology. Variants of the composite board production are stated in Table 3. Test samples for the tensile strength test were produced from these composite boards by means of machining CNC AWJ CT 0806 (computerized numerical control, abrasive water jet technology, Figure 6C). These samples corresponded to the standard ČSN EN ISO 3167 [61]. It was not possible to produce the composite board from FCR 20 by means of vacuum infusion technology. The very small size of FCR20 did not enable penetration of the resin through the layer of filler placed in the mold. The FCR20 behaved as the filtration equipment, eliminating penetration of the matrix (resin). 

AWJ with abrasive grains (garnet MESH 80), traverse speed 250 mm·min^−1^, nozzle diameter 0.8 mm, and working pressure 380 MPa was used for the research. The CNC working parameters were chosen in accordance with the research results on machining of the composite materials, which prevent a delamination of the matrix and the reinforcement [62,63]. Samples were tested on the universal tensile strength testing machine LABTest 5.50ST (a sensing unit AST type KAF 50 kN, evaluating software Test&Motion) equipped with a thermal chamber. The speed of crossbeam motion was 0.6 mm·min^−1^. The setting of the tensile test characteristics was performed in accordance with the standard ČSN EN ISO 527-1 [64]. Testing was performed at a laboratory temperature of 22 ± 2 °C and at increased temperatures 40 °C and 60 °C (see Figure 6D). The test samples were left for 10 min before the testing process itself to equalize the temperatures of the environment and the tested material.

### 2.4. Bonds Reinforced with Composite Layer of Adhesive 

Composite adhesive bonds with 60% of the matrix and 40% of the FCR were the subject of the performed experiments. Already only FCR chemically treated in 5% water solution of NaOH was used within the experiments. The adhesive bonds were prepared in accordance with the standard ČSN EN 1465 [65], i.e., laboratory tests were performed using the standardized test specimens. The structural carbon steel S235J0 was the adhesive bonded material from which test samples of dimensions 100 ± 0.25 × 25 ± 0.25 × 1.6 ± 0.1 mm (Figure 7) determined for the adhesive bonding were cut. Variants of the adhesive bonds production are stated in Table 4. The lapped length of the adhesive layer was 12.5 ± 0.25 mm. 

The adhesive layer thickness was different. It was 108 ± 8 µm at the adhesive bond AB–MHC, 113 ± 11 µm at AB–MHC–FCR20 (5%NaOH), 198 ± 13 µm at AB–MHC–FCR100 (5%NaOH), and 209 ± 16 µm at AB–MHC–FCR700 (5%NaOH).

The adhesive bonded surface of the steel parts was mechanically treated–grit blasted by Garnet MESH 80 and subsequently chemically treated in acetone bath because of degreasing and removing small abrasive particles which could stick to the surface. Roughness parameters were measured with a portable profilometer Mitutoyo Surftest 301. A limit wavelength of the cut-off was set as 0.8 mm. The surface roughness parameters at the grit blasted adhesive bonded material, i.e., structural carbon steel S235J0, were Ra = 1.65 ± 0.18 μm, Rz 10.02 ± 0.65 μm. 

Adhesive bonds were hardened for 72 ± 5 hours at a laboratory temperature of 22 ± 2 °C. The adhesive bonds were fixed with a weight of 750 g. The testing process was performed on universal strength testing machine LABTest 5.50ST (sensing unit AST type KAF 50 kN, evaluating software Test&Motion) at the laboratory temperature. Fracture surface was evaluated in accordance with EN ISO 10365.

The methodology of adhesive bonds testing at determining the strength and elongation at break at shear tensile loading of single lapped adhesive bonds includes determining a reference value of average maximum force at a static tensile test in accordance with the standard ČSN EN 1465 from 10 adhesive bonds by the test speed 0.6 mm·min^−1^ (the reference value was determined from test samples without added filler AB–MHC). Cyclic loading (quasi-static test) of 1000 cycles was conducted with a test speed of 6 mm·min^−1^ between the 5% value, i.e., a bottom limit from the determined reference value of the tensile static test, and 30, 50, and 70%, i.e., an upper limit from the determined reference value; the stamina on the bottom and upper limit was determined to be 0.5 s. The static tensile test was performed after finishing 1000 cycles with the speed of 0.6 mm·min^−1^ until total failure of the adhesive bond.

### 2.5. Evaluation

The evaluation of measured data was performed by means of the program STATISTICA, by ANOVA F-test, i.e., a hypothesis H_0_ presents a statistically insignificant difference among measured data (*p* > 0.05) and a hypothesis H_1_ presents a refusal of the hypothesis H_0_, i.e., there is the statistically significant difference among measured data (*p* < 0.05).

The adhesive bond layer, the interaction of the adhesive layer/adherent (the adhesive bond cut), and the fracture surface were tested by means of the scanning electron microscope TESCAN MIRA 3 GMX SE detector (SEM). The accelerating voltage was 5 to 15 kV, and the working distance was ca. 15 mm. The samples were sputtered with gold by means of the equipment Quorum Q150R ES—sputtering deposition rate using Gold.

## 3. Results and Discussion

Composite boards from microparticles of a particle size of 153 ± 107 µm indicated as FCR 100 and from short fibers of 535 ± 370 µm length indicated as FCR 700 were made by VARTM. It was not possible to make a homogeneous composite board containing the microparticle filler of size 30 ± 20 µm indicated as FCR 20 by vacuum infusion. It did not come to a uniform stratification of FCR 20 at the production. Microparticles worked as filtration equipment, which did not allow MHC to pass. On the contrary, when increasing the power of the vacuum pump, FCR 20 were removed from the mold space.

The test samples made with AWJ technology from the composite boards (Figure 3) were also tested at regarding the increased temperature factor. 

The mechanical properties of MHC and composite systems were characterized by means of tensile strength (Figure 8), the elongation at break (Figure 9), and the modulus of elasticity (Figure 10). The unfilled resin MHC reached a tensile strength if 34 ± 2 MPa at the laboratory temperature. The tensile strength decreased with increasing temperature. The tensile strength fall was 7.8% at the tested temperature 40 °C and up by 26.8% at the temperature of 60 °C. 

The tensile strength was decreased from 1.4% to 31.3% against MHC at the tested temperature of 20 °C by adding FCR 100 and FCR 700 with chemical treatment in 5% water solution of NaOH and also without it. These results correspond to the results of other authors dealing with filling composite materials with the particle filler [14,66]. The results of these authors state the tensile strength decreased depending on the filler size [14,66]. This fall was smaller at the composite materials MHC–FCR100 (5% NaOH) and MHC–FCR700 (5% NaOH), namely from 1.4 to 9.8%. The least significant decrease was at the filler in the form of the short fibers FCR 700 treated in 5% water solution of NaOH.

Composite materials exposed to increased temperatures of 40 and 60 °C showed similar behavior when the tensile strength decrease was up by 45% (MHC–FCR700, 60 °C) against MHC at the 20 °C temperature.

The chemical treatment of FCR in the solution of NaOH had a positive effect on the tensile strength. The negative fall of the tensile strength against the pure adhesive MHC was reduced by the chemical treatment of FCR. The positive effect was at all variants of the experiment, and it moved from 9.1% to 55.1%. More significant improvement occurred at the composite materials filled with short fibers FCR 700 chemically treated in the solution of NaOH, i.e., MHC–FCR700 (5% NaOH). The alkali acting increases the surface structure, which is distinctly visible. It is caused by a removal of natural and artificial impurities from the filler surface [67]. The abovementioned leads to the improvement of the interphase, i.e., of the filler and the matrix [45]. The reason was that the surface microdissolving process increased adhesion among cotton fibers [68].

A difference between FCR without chemical treatment (Figure 11A,C) and with alkali treatment on the FCR surface (Figure 11B,D) is visible from the results of SEM analysis presented in Figure 11. Using alkali treatment in 5% water solution of NaOH for 12 h removed a huge amount of undesirable surface layers from FCR, which was certified by SEM analysis, presented in Figure 11. The tensile strength of the composite systems was increased owing to the performed alkali treatment, which caused an improvement of the interphase between MHC and FCR. Molecules of the matrix might penetrate slightly into the surface of the porous filler. The NaOH treatment improved the wettability of the filler surface by the matrix molecules.

The results of the mechanical tests were also certified by the research on the fracture surfaces by means of SEM, which is visible from Figure 12. A global image of the fracture surface is obvious from Figure 12A, which represents the fracture surface of the composite MHC–FCR700. It is obvious from the figure that it comes down to tearing of FCR from MHC. A low wettability between FCR and MHC decreasing the tensile strength of the composite system (see Figure 8) is evident from Figure 12C. A good wettability of the surface between the matrix and the reinforcement is a basic presumption for successful production of the composite materials [13]. The tensile strength was increased by the chemical treatment of FCR in 5% water solution of NaOH (Figure 8). This result is supported by the findings from the SEM analysis, Figure 12C,E,F. It is evident from Figure 12C,E,F that the adhesive strength between FCR and MHC was higher than the cohesive strength of FCR. Destruction inside the FCR layer is obvious in these figures. A more detailed view is in Figure 12E,F. The MHC based on thermosetting polymer created a brittle fracture, which is evident from Figure 12C. It is visible from Figure 12D that some parts of FCR are of a hollow profile, which could decrease the composite strength. However, the results of Cheung et al. demonstrated that the hollow profile did not decrease the mechanical properties, as was proven at chicken flight feather fibers [69].

The elongation at break was increasing due to increasing temperature at the destructive testing. The most significant increase of the elongation at break occurred at the pure MHC. The elongation at break was 3% at MHC at the laboratory temperature. Owing to the increased temperature, the elongation at break was increased, namely to 165% at 40 °C and up to 226% at 60 °C. Adding FCR led to the decrease of the elongation at break increase, both at the laboratory temperature of 20 °C and at increased testing temperatures, i.e., 40 and 60 °C, which is evident from Figure 9. The alkali treatment in NaOH had a positive effect, namely at the temperature 20 °C, when a mild increase of the elongation at break to the pure MHC occurred at the composite material reinforced with the cotton filler in the form of short fibers, indicated as MHC–FCR700 (5% NaOH). This potential decreased with increasing temperature. 

The elongation at break increase is in accordance with analogous results, e.g., with PP/coir fiber composites treated in NaOH, where the increase of the elongation at break was also observed [70]. 

Often bad wettability by the matrix, which usually decreases the tensile strength, belongs among the significant disadvantages of the natural filler utilization in the area of the polymeric composites [17,71,72,73]. Chemical treatment in 5% water solution of NaOH minimalizes this negative factor having an influence on the tensile strength.

The modulus of elasticity of tested materials is visible from Figure 10. It is obvious from the results that the modulus of elasticity was increased by adding FCR, namely FCR100 and FCR700 without chemical treatment. A significant decrease of the modulus of elasticity owing to the temperature at the stress test is also obvious from the results [74]. When increasing the temperature at the polymeric materials, the modulus of elasticity and the tensile strength decrease, and the elongation at break increases [74]. 

The reinforcing optimization of MHC matrices with FCR fillers depends on the interphase interaction, i.e., the stress transfer comes through the adhesion.

The results of the statistical testing are visible in Table 5. The statistical testing proved significant differences in the tensile strength of the tested materials, i.e., MHC and composites depending on the temperature change. The composite material MHC–FCR100 was the only exception at which the statistically significant difference depending on the tested temperatures 20 °C, 40 °C, and 60 °C was not proven. The statistically significant difference at the elongation at break depending on the temperature change (interval from 20 °C to 60 °C) at the destructive testing was proven at all variants of the experiment, i.e., tested MHC materials and composites. There is no statistical difference between 40 °C and 60 °C for MHC–FCR100 (*p* = 0.8145) and MHC–FCR100 (5% NaOH), where *p* = 0.9137.

Conclusions that adding filler into thermosetting epoxy-based matrix affects (improves) the mechanical properties of the resulting composite were certified [30,31].

It cannot be assumed that the adhesive bonds will keep a significant bearing capacity during their whole service life [75,76]. Service life conditions usually comprise the cyclic stress, i.e., cyclic fatigue. This cyclic fatigue causes damages to the adhesive bonds which are irreversible. This process also influences structural adhesive bonds at relatively small loading values as a consequence of the delamination of the adhesive layer and the adhesive bonded material, which influences the adhesive bonds service life in a negative way [77]. Further, these small loading values, which repeat themselves (the cyclic degradation), decrease the static strength and the fatigue service life of the adhesive bonds [75]. Thus, the tests of the adhesive bonds cyclic loading, which belongs among the important aspects of the practical application of the adhesive bonds, are essential from that reason [77,78,79]. 

The adhesive bond strength reference value adjustment necessary to determine the maximum force for the quasi-static test proved the influence of adding FCR. It is obvious from Figure 13 that the adhesive bond strength decreased with increasing size of FCR, namely up of 22% at AB–MHC–FCR700 (5% NaOH). A mild increase of the adhesive bond strength of 9.4% occurred at the adhesive bonded layer reinforced with FCR 20. A mild increase of the adhesive bond strength was proven with the use of very small filler in the form of microparticles [80]. 

It was certified based on the experiment results that the epoxy adhesives also keep their strength at a higher volume of the filler [30,31]. 

A similar behavior of the adhesive bonds was also proven at the elongation at break (Figure 14). The increase of the elongation at break of 56% occurred towards the adhesive bonds using the pure adhesive (AB–MHC). The elongation at break decreased by 14% at the FCR 100 and by 36% at FCR 700. The negative effect of the increasing size of the filler was proven again in the area of the adhesive bonds. In terms of statistical testing, the difference in various variants of the adhesive bonds in respect of the adhesive bond strength (*p* = 0.000) and adhesive bond elongation at break (*p* = 0.000) was proven.

The fracture surface was adhesive and adhesive–cohesive at adhesive bonds. The results are presented in Table 6, from which the difference between AB–MHC and AB–MHC with FCR is not obvious. 

The results of quasi-static tests of adhesive bonds at 1000 cycles are visible in Figure 15 and Figure 16 and Table 7. The deformation almost did not occur at the quasi-static tests at low values, i.e., the cyclic tests in the interval from 5% to 30% at the loading force from 208 to 1246 N and from 5% to 50% at the loading force from 208 to 2076 N; the adhesive layer did not change—it did not come to deformation. The situation is different at the cyclic loading values in the interval from 5% to 70% at the loading corresponding to the interval 208 to 2906 N. It comes to the significant viscoelastic behavior of the adhesive layer, i.e., the creep, between the first and the last cycle; the deformation inside the adhesive layer occurs. The viscoelastic behavior of the adhesive layer is obvious from Table 8, from which increasing influence depending on increasing values of the loading force at the quasi-static test, i.e., 30%, 50%, and 70%, is evident. This influence was certified not only at the testing itself but also at the final deformation of the adhesive bond after the last cycle. 

Further, the influence of the FCR was shown as essential. The results of the quasi-static tests proved that the FCR 20 is suitable. This filler withstood 800 cycles on average at the cyclic loading in the interval from 5% to 70% at the loading corresponding to the interval 208 to 2906 N. It was the best result compared to adhesive bonds AB–MHC and AB–MHC–FCR 100 and 700. FCR 700 (short-fiber adhesive layer) seemed to be the least suitable in terms of the cyclic loading resistance. An irregular shape of short randomly oriented fibers is a reason of this effect. The resulting number of the cycles at the single quasi-static tests is presented in Table 8.

All tested adhesive bonds exposed to the quasi-static tests, i.e., to the cyclic tests in the interval from 5% to 30% at the loading force from 208 to 1246 N and from 5% to 50% at the loading force from 208 to 2076 N, withstood the upper limit of the tests, i.e., 1000 cycles. The situation is different at the adhesive bonds exposed to the cyclic loading values in the interval from 5% to 70% at the loading corresponding to the interval 208 to 2906 N. The number of finished cycles differed significantly. At the adhesive bonds AB–MHC and AB–MHC–FCR 20 and 100, 3 to 4 adhesive bonds from the total series 10 pieces finished the tests. At the adhesive bond AB–MHC–FCR 700, the 1000th cycle was finished at no adhesive bonds. The experiment results certified the presumption that repeated cyclic loading of the adhesive bonds with higher values of the loading force could lead to premature failure of the adhesive bond at a relatively small number of the cycles [75].

An exhibition of quasi-static curves for the adhesive bonds with layer indicated as AB–MHC–FCR 20 is in Figure 17, Figure 18, Figure 19 and Figure 20. It is possible to compare the low cyclic test from 5% to 30% (208–1246 N), 5% to 50% (208–2076 N) and 5% to 70% (208–2906 N) of the adhesive bond AB–MHC–FCR 20. It is possible to compare the low cyclic test with the highest tested loading, i.e., from 5% to 70% (208–2906 N) of the adhesive bond AB–MHC–FCR 20 from Figure 19 and Figure 20. It is evident from Figure 17, Figure 18 and Figure 19 that the first 1000 cycles were performed, and then, immediately, the test was performed to break (without removing the sample from the machine between the last cycle and the test until break). Figure 19 presents the finishing of the 1000 cycles and Figure 20 the finishing of 343 cycles from the 1000.

The fracture surface of the adhesive bonds after finishing low-cyclic tests was of a combined type, i.e., adhesive–cohesive (ACF) and of a pure adhesive type (AF). The influence of the quasi-static test is not obvious from the results presented in Table 6. The exhibition of the adhesive–cohesive fracture surface is presented in Figure 21. A brittle fracture inside the adhesive layer is evident from Figure 21B. A destruction of the FCR 700, due to which the cohesive failure of the adhesive layer occurs, is evident from Figure 21C. 

An exhibition of the adhesive bonds cut is evident from Figure 22. Figures present the influence of the interaction of the adhesive bonded material, the FCR, and the MHC. Figure 22A presents the global view on the cut through the adhesive bond AB–MHC–FCR 20 (5% NaOH), which was not exposed to the cyclic loading.

A good wettability of the adhesive layer and the adhesive bonded material is a basic success of the adhesive bonding technology usage in practice [14,76,81]. Microscopic cracks inside the adhesive layer caused by the cyclic loading or owing to the changes of the temperature can be eliminated by the usage of the filler [76]. 

A good interaction between adhesive bonded layer consisted of FCR and MHC, and the adhesive bonded material is visible in Figure 22A. A homogeneity of the FCR 700 is obvious from Figure 22B. Smaller interaction between the adhesive bonded material and MHC with FCR owing to the cyclic loading at the quasi-static test from 50% to 30% is also evident. This delamination leads to the initiation of the adhesive–cohesive fracture surface. The delamination is visible in the right upper part of the cut through the adhesive bond (Figure 22B). Similar delamination between the adhesive bonded material and the adhesive layer is visible in Figure 22C. This delamination causes adhesive failure of the adhesive bond. The small cracks are a consequence of the dynamic tests with higher loading during testing, i.e., from 5% to 70% (208–2906 N) and in the cases of smaller loading values from 5% to 30% (208–1246 N) and 5% to 50% (208–2076 N) of finishing 1000 cycles. 

The results of the statistical testing are visible in Table 8. The statistical testing proved significant differences in the strength and the deformation of the adhesive bond among tested adhesive bonds, i.e., AB–MHC and adhesive bonds with composite layer MHC–FCR depending on the type of the quasi-static test. 

Generally, it can be said that the interest in composites filled with the biological reinforcing phase in the form of particles and fibers is increasing [82]. The low price, easy availability, and renewability of these fillers is the reason according to Danyadi and Renner and Ruggiero et al. [14,82].

## 4. Conclusions

This paper evaluated the utilization possibilities of the microparticle and short-fiber filler based on cotton post-harvest line residues in the area of the polymeric composite materials. The study builds on the utilization of natural fillers in the area of composite materials. The results proved that the utilization of Turkish cotton post-harvest line residues is possible in the area of the polymeric composite, and it brings a possibility of effective material utilization of this minor agricultural commodity. This material utilization provides another alternative of by-product utilization for farmers dealing with cotton growing. The filler for the composite materials in the form of particles and short fibers can be quickly and in a relatively effective way prepared from cotton growing residues. The potential use of these composites is, namely, in the area of adhesive bonds filled with biological filler where it is not possible to define exactly the stress type and its direction in advance.

Essential research conclusions:SEM analysis of the FCR based on cotton post-harvest line residues proved the influence of alkali treatment in the water solution of NaOH on the surface structure of fibers, i.e., removal of surface layers. Further, the positive influence of FCR alkali treatment on the wettability with MHC was certified.Composite materials: The tensile strength and modulus of elasticity of the composite materials decreased owing to adding FCR. The composite materials with FCR, whose surface was treated in 5% water solution of NaOH, showed a smaller fall of the tensile strength towards MHC. The chemical treatment of FCR in the solution of NaOH showed up in a positive way on the tensile strength. The negative fall of the tensile strength against the pure adhesive MHC managed to be reduced by the chemical treatment of FCR. The FCR 700, i.e., the short-fiber filler, was the best in terms of the application of FCR in composite materials (compared to the particle filler). This conclusion was also certified at increased tested temperatures of 40 °C and 60 °C. Composite materials showed similar behavior also in terms of the tested elongation at break.Adhesive bonds: The strength of the adhesive bond decreased with increasing size of FCR at the adhesive bonds, except for the adhesive bonds with the adhesive layer reinforced with FCR 20, where a mild increase of the adhesive bond strength occurred. Results of the quasi-static tests certified that the FCR 20 is suitable not only at the static test but also at the cyclic loading. Viscoelastic properties of the adhesive (specifically creep) significantly showed themselves at the cyclic loading values in the interval from 5% to 70% at the loading corresponding to the interval of 208 to 2906 N, whereas these viscoelastic properties almost did not show themselves at low values of the cyclic tests at the interval from 5% to 30% at the loading force from 208 to 1246 N and from 5% to 50% at the loading force from 208 to 2076 N.

## Figures and Tables

**Figure 1 polymers-11-01106-f001:**
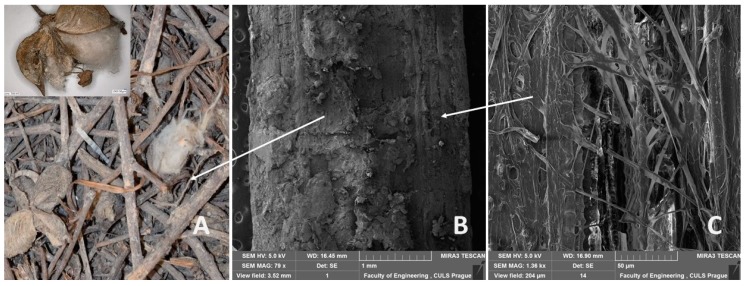
Cotton post-harvest line residues: (**A**): general view on residues, (**B**): SEM image of stem surface (MAG 79 x), (**C**): SEM image of detailed stem surface structure (MAG 1.36 kx).

**Figure 2 polymers-11-01106-f002:**
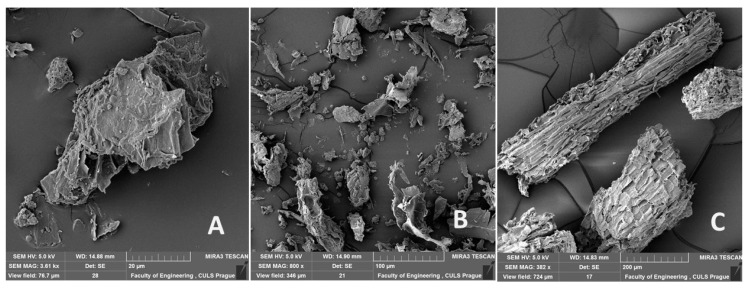
SEM images of reinforced cotton filler (FCR): (**A**): FCR based on cotton post-harvest line microparticles 20 µm (MAG 3.61 kx), (**B**): FCR based on cotton post-harvest line microparticles 100 µm (MAG 800 x), (**C**): FCR with short fibers of length 700 µm (MAG 382 x).

**Figure 3 polymers-11-01106-f003:**
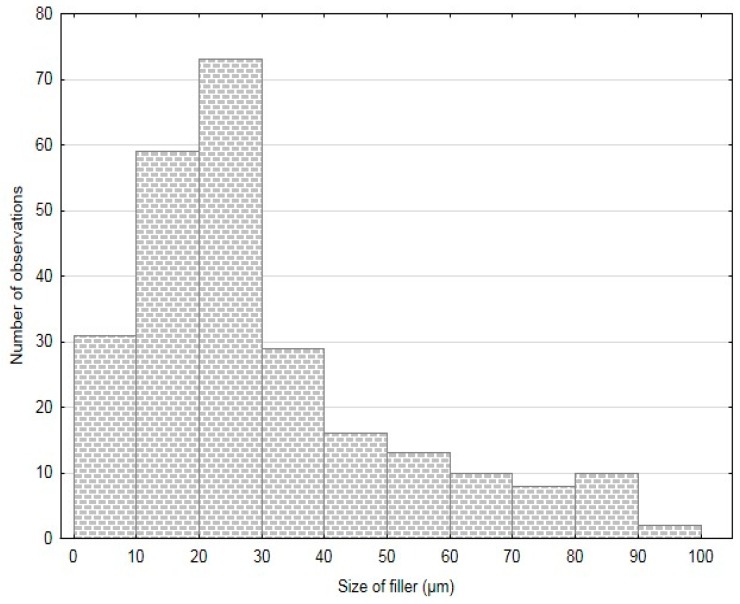
Histogram of FCR 20.

**Figure 4 polymers-11-01106-f004:**
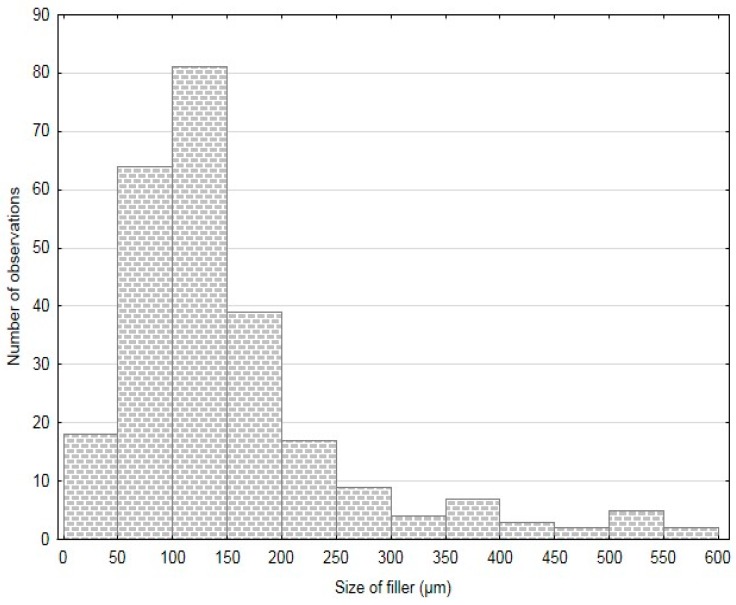
Histogram of FCR 100.

**Figure 5 polymers-11-01106-f005:**
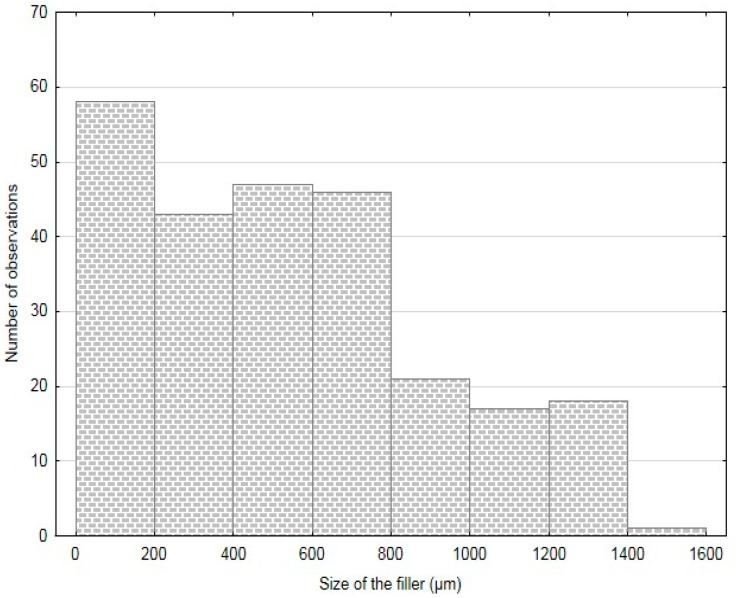
Histogram of FCR 700.

**Figure 6 polymers-11-01106-f006:**
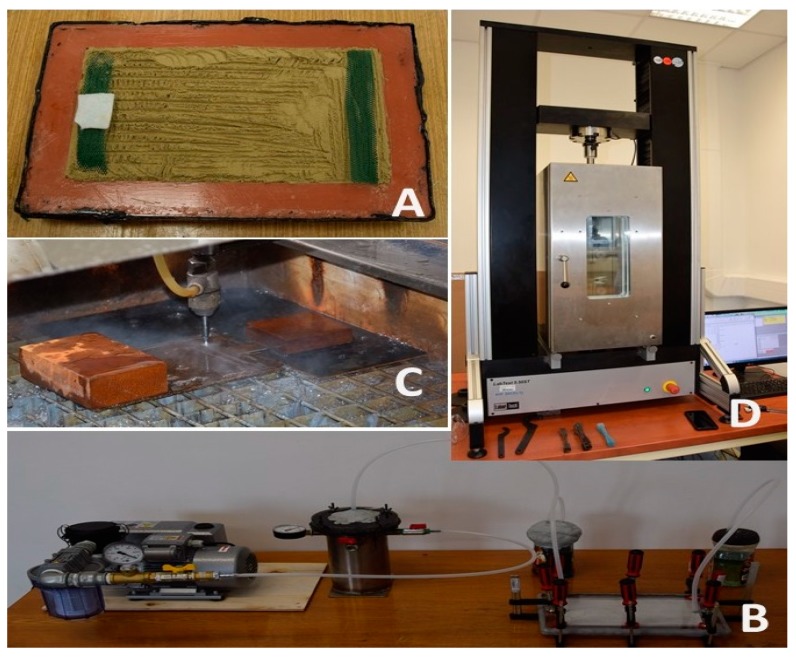
Testing: (**A**): mold with FCR, (**B**): production of composite board by vacuum infusion, (**C**): production of test samples—AWJ technology, (**D**): testing on universal testing machine with thermal chamber.

**Figure 7 polymers-11-01106-f007:**
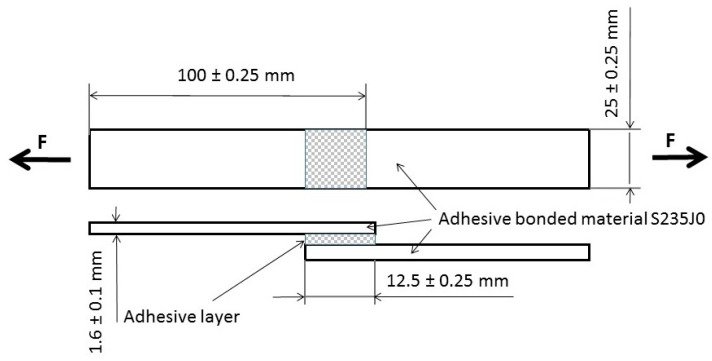
Schema of test sample according to ČSN EN 1465.

**Figure 8 polymers-11-01106-f008:**
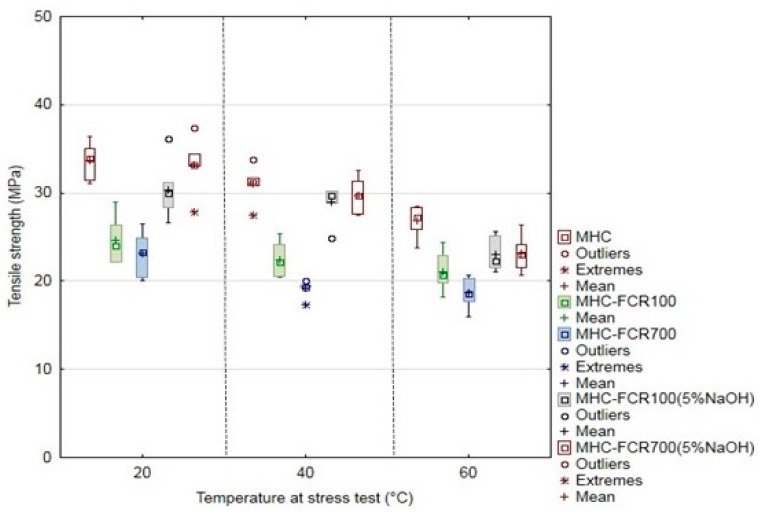
Tensile strength of Havel Composites matrix (MHC) and composite materials MHC–FCR.

**Figure 9 polymers-11-01106-f009:**
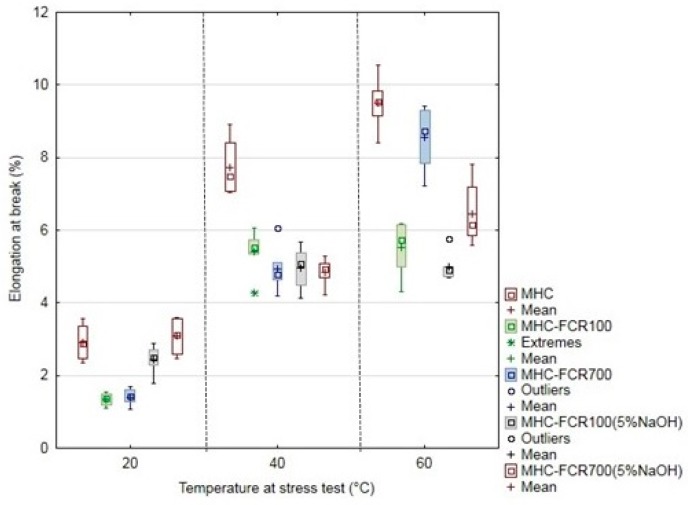
Elongation at break of MHC and composite materials MHC–FCR.

**Figure 10 polymers-11-01106-f010:**
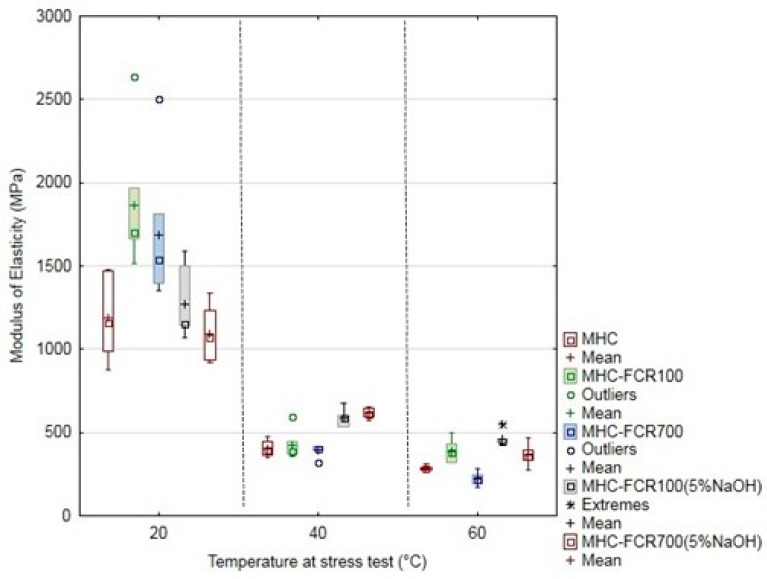
Modulus of elasticity of MHC and composite materials MHC–FCR.

**Figure 11 polymers-11-01106-f011:**
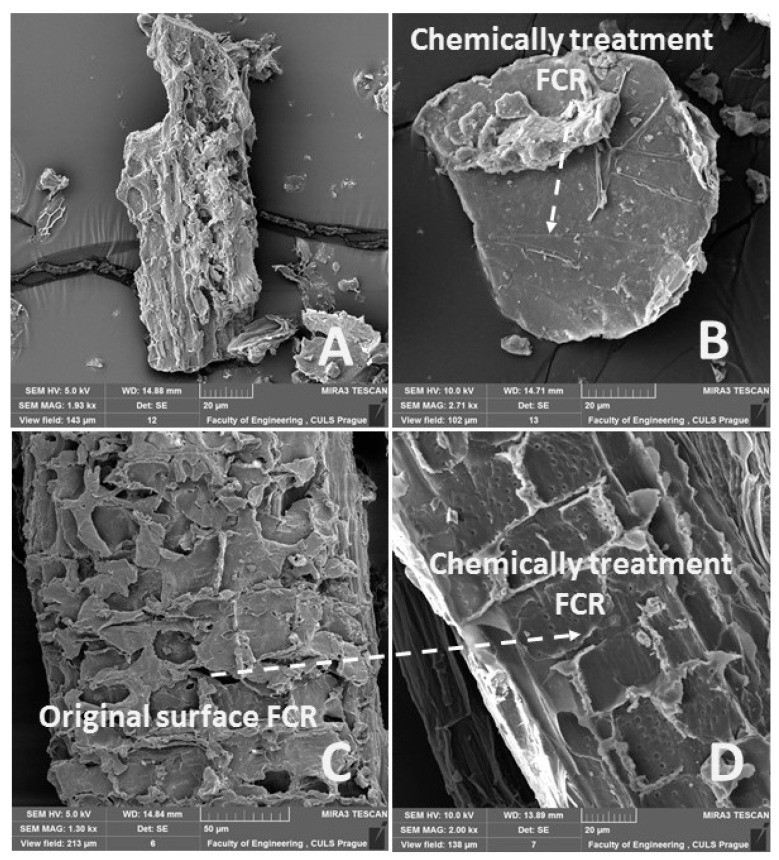
SEM images of FCR: (**A**): reinforced cotton filler of particle size indicated as FCR 100 (MAG 1.93 kx), (**B**): reinforced cotton filler of particle size indicated as FCR 100 chemically treated in 5% solution of NaOH for 12 h (MAG 2.71 kx), (**C**): reinforced cotton filler from short fibers indicated as FCR 700 (MAG 1.30 kx), (**D**): reinforced cotton filler from short fibers indicated as FCR 700 chemically treated in 5% solution of NaOH for 12 h (MAG 2.00 kx).

**Figure 12 polymers-11-01106-f012:**
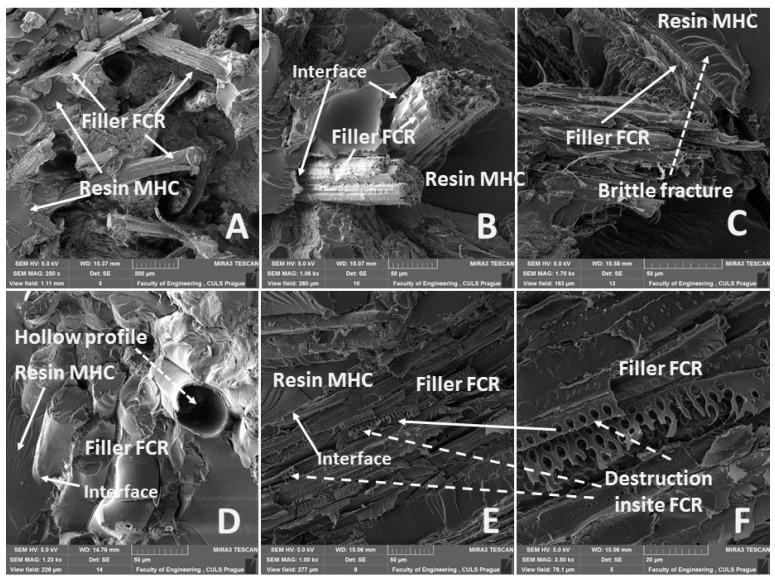
SEM images of fracture surface—static tensile test of composite materials: (**A**): MHC–FCR700 20 °C (MAG 250 x), (**B**): MHC–FCR100 40 °C (MAG 1.06 kx), (**C**): MHC–FCR100 (5% NaOH) 20 °C (MAG 1.70 kx), (**D**): MHC–FCR700 (5% NaOH) 60 °C (MAG 1.23 kx), (**E**): MHC–FCR700 (5% NaOH) 40 °C (MAG 1.00 kx), (**F**): destruction of FCR in composite MHC–FCR700 (5% NaOH) 40 °C (MAG 3.50 kx).

**Figure 13 polymers-11-01106-f013:**
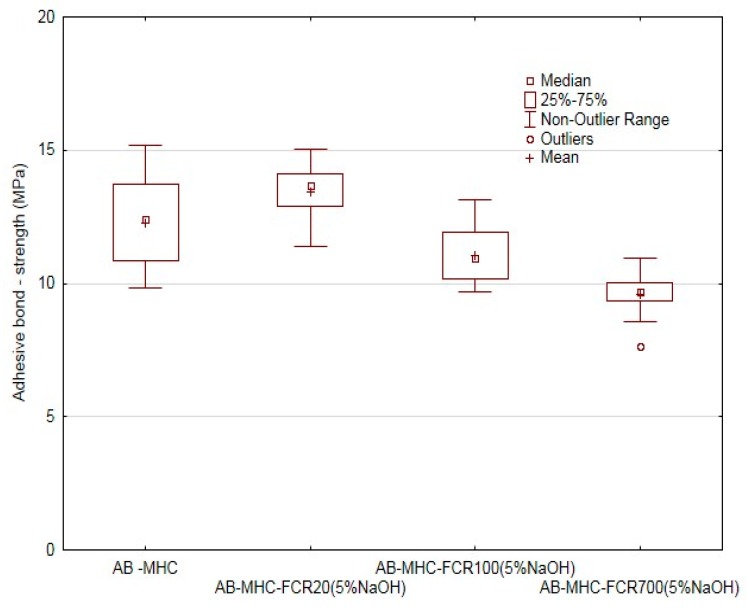
Static adhesive bond test—adhesive bond strength.

**Figure 14 polymers-11-01106-f014:**
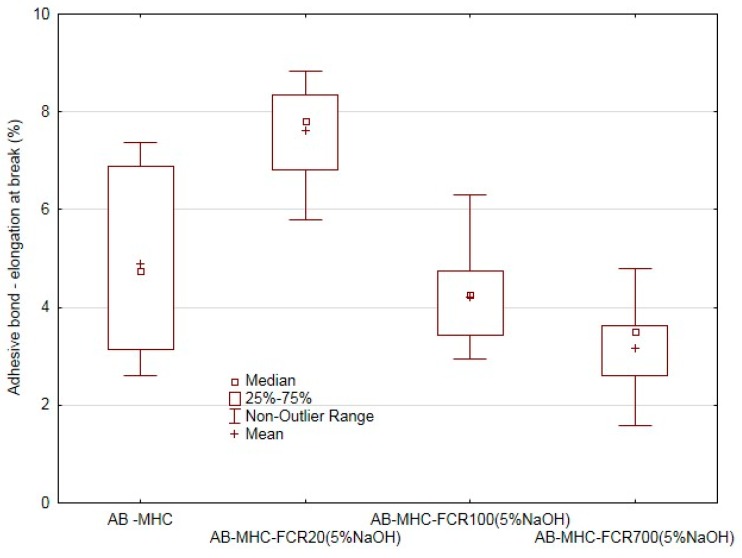
Static adhesive bond test—elongation at break of adhesive bond.

**Figure 15 polymers-11-01106-f015:**
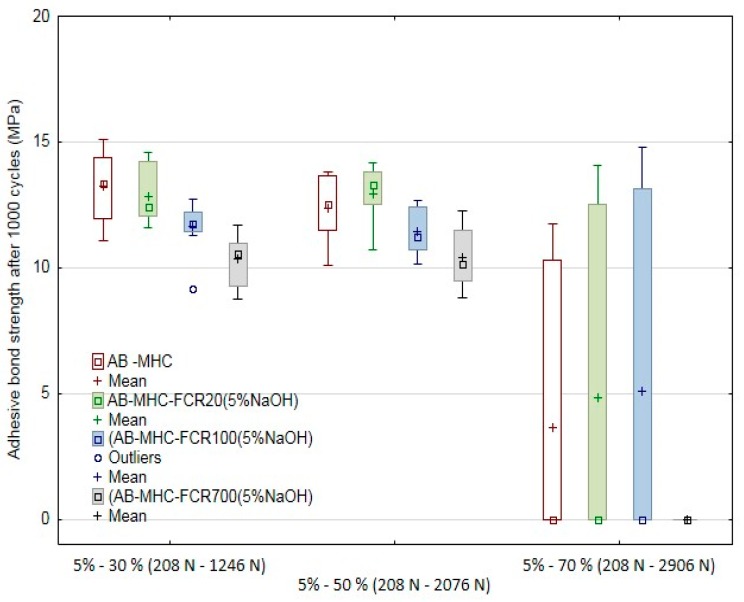
Quasi-static test of adhesive bonds after 1000 cycles—adhesive bond strength.

**Figure 16 polymers-11-01106-f016:**
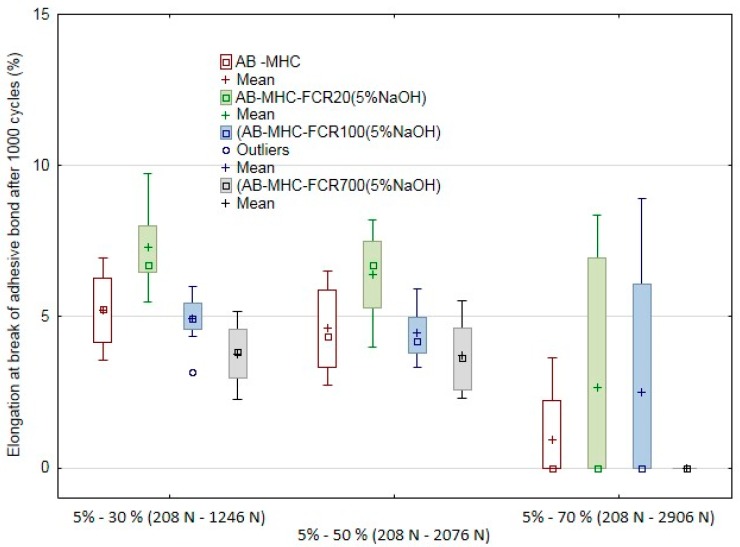
Quasi-static test of adhesive bonds after 1000 cycles—elongation at break of adhesive bonds.

**Figure 17 polymers-11-01106-f017:**
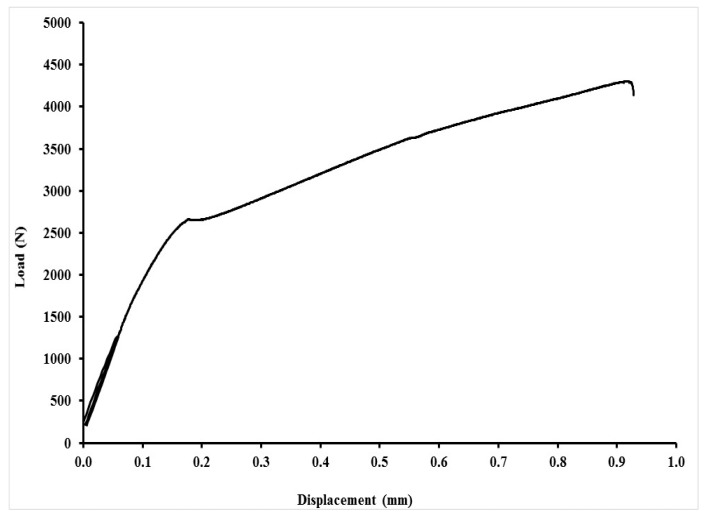
Low-cyclic test from 5% to 30% (208–1246 N) of adhesive bond AB–MHC–FCR 20 (5% NaOH).

**Figure 18 polymers-11-01106-f018:**
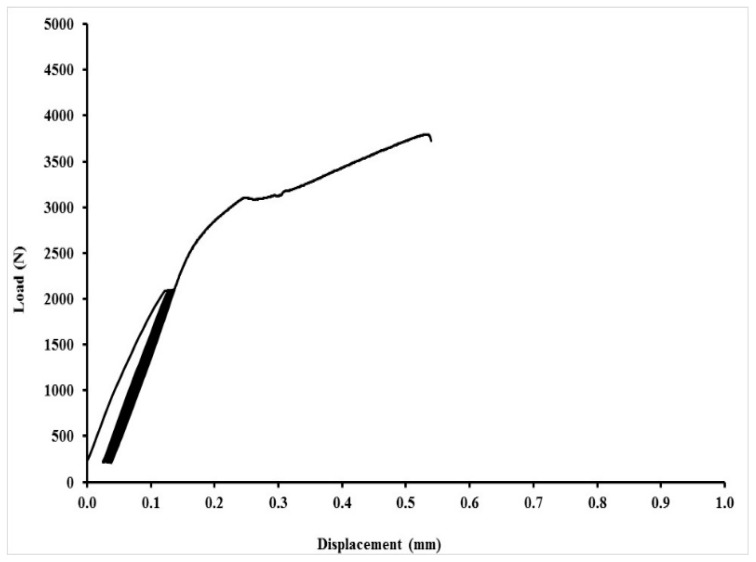
Low-cyclic test from 5% to 50% (208–2076 N) of adhesive bond AB–MHC–FCR 20 (5% NaOH).

**Figure 19 polymers-11-01106-f019:**
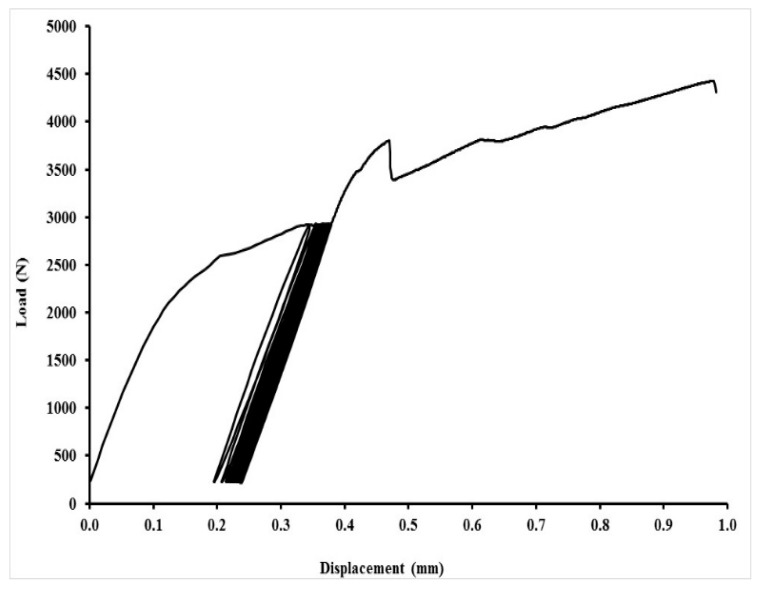
Low-cyclic test from 5% to 70% (208–2906 N) of adhesive bond AB–MHC–FCR 20 (5% NaOH)—1000 cycles.

**Figure 20 polymers-11-01106-f020:**
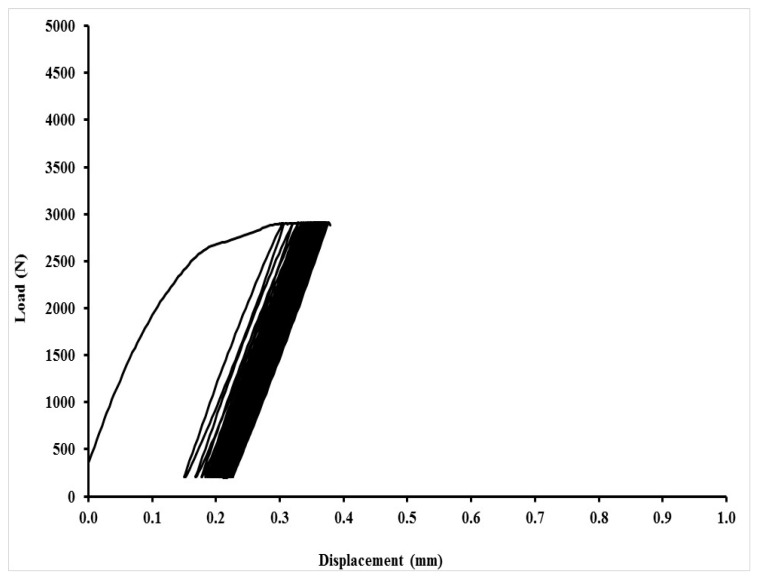
Low-cyclic test from 5% to 70% (208–2906 N) of adhesive bond AB–MHC–FCR 20 (5% NaOH)—343rd cycle finished (premature destruction of adhesive bond).

**Figure 21 polymers-11-01106-f021:**
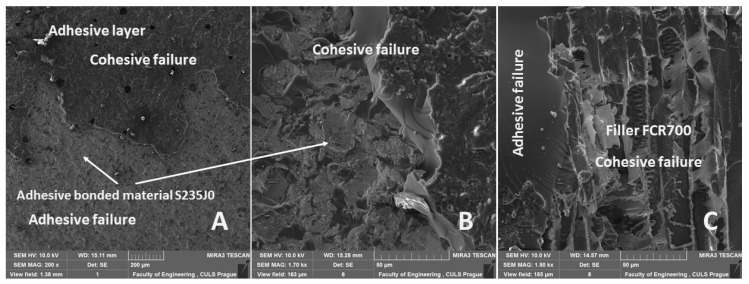
SEM images of fracture surface of adhesive bond AB–MHC at low cyclic test from 5% to 70%: (**A**): adhesive–cohesive fracture surface (MAG 200 x), (**B**): detailed view on interphase of adhesive–cohesive fracture surface (MAG 1.70 kx), (**C**): detailed view on cohesive failure of interphase MHC and FCR 700 (MAG 1.50 kx).

**Figure 22 polymers-11-01106-f022:**
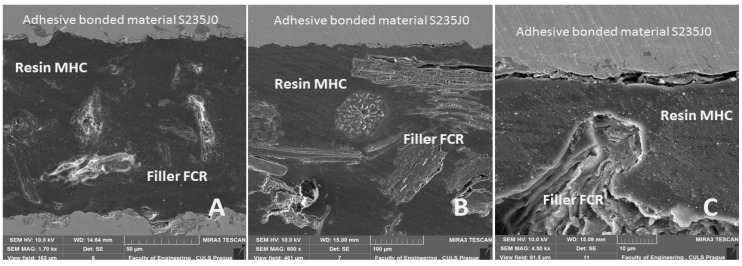
SEM images of cut through adhesive bond: (**A**): cut through adhesive bond AB–MHC–FCR20 (5% NaOH) without cyclic loading (MAG 1.70 kx), (**B**): cut through adhesive bond AB–MHC–FCR700 (5% NaOH) exposed to cyclic loading at quasi-static test from 5% to 30% (MAG 600 x), (**C**): cut through adhesive bond AB–MHC–FCR700 exposed to cyclic loading at quasi-static test from 5% to 70% (MAG 4.50 kx).

**Table 1 polymers-11-01106-t001:** Measuring of used FCR.

Indication	Filler Type	Arithmetic Mean (µm)	Mode (µm)	Median (µm)
FCR 20	Particle	30	21	25
FCR 100	Particle	153	93	123
FCR 700	Short fibres	535	711	509

**Table 2 polymers-11-01106-t002:** Properties of resin/hardener—technical sheet according to producer [59,60].

Resin Property	Resin	Hardener
Epoxy mass equivalent (g·mol^−1)^	180–196	41
Epoxy index (mol·1000g^−1^)	0.51–0.56	-
Flash-point	>150	-
Viscosity (mPa·s^−1^ at 25 °C)	500–900	4-15
Density (g·cm^−3^)	1.12–1.16	0.91–0.94

**Table 3 polymers-11-01106-t003:** Indication of two-component epoxy adhesive modification at production of composite material.

Composite Material	Indication Characteristics
MHC	Matrix Havel Composite, no filler
MHC–FCR100	Matrix Havel Composite, reinforced cotton filler of particle size 100 µm (modus)
MHC–FCR700	Matrix Havel Composite—reinforced cotton filler of short fibre length 700 µm (modus)
MHC–FCR100 (5% NaOH)	Matrix Havel Composite—reinforced cotton filler of particle size 100 µm (modus) chemically treated in 5% water solution of NaOH for time 12 h.
MHC–FCR700 (5% NaOH)	Matrix Havel Composite—reinforced cotton filler of short fibre length 700 µm (modus) chemically treated in 5% water solution of NaOH for time 12 h

**Table 4 polymers-11-01106-t004:** Indication of two-component epoxy adhesive modification at structural adhesive bonds production.

Adhesive bonds	Indication Characteristics
AB–MHC	Adhesive bond–Matrix Havel Composite, no filler
AB–MHC–FCR20 (5% NaOH)	Adhesive bond–Matrix Havel Composite—reinforced cotton filler of particle size 20 µm (modus) chemically treated in 5% solution of NaOH for time 12 h
AB–MHC–FCR100 (5% NaOH)	Adhesive bond–Matrix Havel Composite—reinforced cotton filler of particle size 100 µm (modus) chemically treated in 5% solution of NaOH for time 12 h
AB–MHC–FCR700 (5% NaOH)	Adhesive bond–Matrix Havel Composite—reinforced cotton filler of short fiber length 700 µm (modus) chemically treated in 5% solution of NaOH for time 12 h

**Table 5 polymers-11-01106-t005:** Statistical evaluation of static tensile test according ANOVA F-test with stated parameter *p* in significance level α 0.05 at change of tested temperature.

Tensile Test	MHC	MHC–FCR100	MHC–FCR700	MHC–FCR100 (5% NaOH)	MHC–FCR700 (5% NaOH)
Tensile strength (MPa)	0.0001	**0.0584**	0.0016	0.0003	0.0001
Elongation at break (%)	0.0000	0.0000	0.0000	0.0000	0.0000

**Table 6 polymers-11-01106-t006:** Fracture surface evaluation according to EN ISO 10365 (AF—adhesive failure, ACF—adhesive–cohesive failure, CF—cohesive failure).

Adhesive Bond	Characteristics of Adhesive Bond Test	AF	ACF	CF
AB–MHC	Standard adhesive bond test	6	4	0
	Quasi-static test from 5% to 30% (208–1246 N)	1	9	0
	Quasi-static test from 5% to 50% (208–2076 N)	3	7	0
	Quasi-static test from 5% to 70% (208–2906 N)	1	9	0
AB–MHC–FCR20 (5% NaOH)	Standard adhesive bond test	5	5	0
	Quasi-static test from 5% to 30% (208–1246 N)	6	4	0
	Quasi-static test from 5% to 50% (208–2076 N)	5	5	0
	Quasi-static test from 5% to 70% (208–2906 N)	1	9	0
AB–MHC–FCR100 (5% NaOH)	Standard adhesive bond test	1	9	0
	Quasi-static test from 5% to 30% (208–1246 N)	5	5	0
	Quasi-static test from 5% to 50% (208–2076 N)	3	7	0
	Quasi-static test from 5% to 70% (208–2906 N)	5	5	0
AB–MHC–FCR700 (5% NaOH)	Standard adhesive bond test	3	7	0
	Quasi-static test from 5% to 30% (208–1246 N)	8	2	0
	Quasi-static test from 5% to 50% (208–2076 N)	5	5	0
	Quasi-static test from 5% to 70% (208–2906 N)	1	9	0

**Table 7 polymers-11-01106-t007:** Results of quasi-static test of adhesive bond.

Characteristics of Adhesive Bond (AB)	Quasi-Static Test	Number of Cycles	Number of Test Samples (Number of Finished Cycles/Total Number of Tests)	Relative Deformation after Finishing 1st Cycle	Relative Deformation after Last Cycle
AB–MHC	from 5% to 30% (208–1246 N)	1000 ± 0	10/10	0.09%	0.09%
	from 5% to 50% (208–2076 N)	1000 ± 0	10/10	0.12%	0.18%
	from 5% to 70% (208–2906 N)	632.4 ± 268.9	3/10	0.43%	0.75%
AB–MHC–FCR20 (5% NaOH)	from 5% to 30% (208–1246 N)	1000 ± 0	10/10	0.02%	0.05%
	from 5% to 50% (208–2076 N)	1000 ± 0	10/10	0.12%	0.20%
	from 5% to 70% (208–2906 N)	800 ± 210	4/10	1.51%	1.96%
AB–MHC–FCR100 (5% NaOH)	from 5% to 30% (208–1246 N)	1000 ± 0	10/10	0.05%	0.06%
	from 5% to 50% (208–2076 N)	1000 ± 0	10/10	0.15%	0.23%
	from 5% to 70% (208–2906 N)	606 ± 353	4/10	0.81%	1.31%
AB–MHC–FCR700 (5% NaOH)	from 5% to 30% (208–1246 N)	1000 ± 0	10/10	0.06%	0.06%
	from 5% to 50% (208–2076 N)	1000 ± 0	10/10	0.16%	0.25%
	from 5% to 70% (208–2906 N)	155 ± 137	0/10	0.74%	1.29%

**Table 8 polymers-11-01106-t008:** Statistical evaluation of adhesive bond at different low-cyclic fatigue values (from 5% to 70% (208–2906 N), from 5% to 50% (208–2076 N) and from 5% to 30% (208–1246 N)), according to ANOVA F-test with presented parameter *p* in significance level α 0.05.

Testing of Adhesive Bonds at Shear Tensile Stress	AB–MHC	AB–MHC–FCR20	AB–MHC–FCR100	AB–MHC–FCR700
Adhesive bond strength (MPa)	0.0001	0.0005	0.0010	0.0000
Elongation at break of adhesive bond (%)	0.0000	0.0005	0.0353	0.0000

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
