# Peer review of "Material Utilization of Cotton Post-Harvest Line Residues in Polymeric Composites"

_polymers, 2019, doi:10.3390/polym11071106_

Reviewer 1 Report

Important topic, utilizing renewable and waste materials is interesting.

Check the language, sometimes misleading or obscure phrases are used. I believe I got the meaning, but some words/phrases are just wrong.

It would be good to also to further discuss issues regarding hygroscopicity of natural fibers vs. man made fibers.

Comments about the text (L denotes Line):

L3: Material utilization of cotton post-harvest line residues in area of polymeric composites, word area is not needed

L31 explain somewhere why this is hybrid

L42: “Renewable resources are materials” Rephrase

L53 are e.g. jute, flax and hemp really vegetables?

L57 “Natural resources benefit in the area of fibre and particle composite material fillers..” obscure, rewrite

L60 “..leads to a decrease of the mass and the costs of single..” repetition to L45

L94 – have you studied other treatments like corona or plasma?

L117 interfaces between the matrix and fiber could also be mentionted

L134 ‘at the same time.’ – how where they analyzed simultaneously?

L143 not sure if description of formation of cellulose is required

L149 this is mentioned in L132, should be only in one place

L193 ‘..(MHC) was the matrix [1,2].’ What are the references for?

L200 you could probably use the term VARTM for the method

L211 You don’t need to explain what vacuum pump is and does

L222 ‘The CNC setting..” I think setting is not the best term to be used here

L276 ‘..dusted with gold..’ sputtered might be a better term

L283-285 Interesting finding!

L348 “..matrix and a hydrophilic, which usually..” rephrase

L370 Interesting, any explanation why the MHC-FCR100 has so good thermal stability?

L450 Displacement in percentage in the Figures 15-18 would be more informative.

L522 Again, why hybrid?

Author Response

Review 1

Changes in the text according to the reviewer 1 are made by green colour.

L3: Material utilization of cotton post-harvest line residues in area of polymeric composites, word area is not needed

Corrected.

L31 explain somewhere why this is hybrid

Added into the text. Hybrid – a term used at adhesive bonds where the adhesive contains filler, namely biological.

L42: “Renewable resources are materials” Rephrase

Corrected.

L53 are e.g. jute, flax and hemp really vegetables?

They are vegetable fibres.

L57 “Natural resources benefit in the area of fibre and particle composite material fillers..” obscure, rewrite

Rewritten.

L60 “..leads to a decrease of the mass and the costs of single..” repetition to L45

Corrected – omitted in L45.

L94 – have you studied other treatments like corona or plasma?

Yes, but in another research. Plasma have turned out to be effective but expensive. It is not suitable for the practice. Therefore it is not in other researches.

L117 interfaces between the matrix and fiber could also be mentionted

Corrected.

L134 ‘at the same time.’ – how where they analyzed simultaneously?

Corrected.

L143 not sure if description of formation of cellulose is required

I agree. It is omitted.

L149 this is mentioned in L132, should be only in one place

It is omitted in L132.

L193 ‘..(MHC) was the matrix [1,2].’ What are the references for?

Thank you. The mistake was made when editing Mendeley.

L200 you could probably use the term VARTM for the method

Added.

L211 You don’t need to explain what vacuum pump is and does

I agree. The part describing the principle was omitted.

L222 ‘The CNC setting..” I think setting is not the best term to be used here

Corrected.

L276 ‘..dusted with gold..’ sputtered might be a better term

Corrected.

L283-285 Interesting finding!

Yes, we were also surprised. It was repeatedly certified also with another small filler.

L348 “..matrix and a hydrophilic, which usually..” rephrase

Corrected.

L370 Interesting, any explanation why the MHC-FCR100 has so good thermal stability?

We do not know. We will further investigate it with colleagues from other workplaces.

L450 Displacement in percentage in the Figures 15-18 would be more informative.

Added into text. Also other reviewer asked for a correction.

L522 Again, why hybrid?

Removed.

Reviewer 2 Report

See attached file.

Paper was very tedious to read. This must be improved.

Author Response

Review 2

Changes in the text according to the reviewer 2 are made by red colour. I thank the reviewer for detailed analysis of the paper.

Line

Comments

33

Explain also in Summary what FCR is.

Added.

44-46

Reformulate, second part of sentence is   unclear

Second   part of the sentence was omitted. It is not essential.

46

What means “secondary products”

This   term is used, in waste area, for the material arisen from waste.   

50

Reformulate

Corrected.

68

The term “biocomposite” is only allowed   of the whole material is based on natural resources, not only the filler.

I   agree, I apologize.  It had been added based   on a recommendation of one author. It was deleted also in next text.   Nevertheless this term can be found also in presented connection.  

97

Biocomposites also need the matrix from   renewable resources, not only the filler. I agree, it is logical. Nevertheless this term can be   found also in presented connection.  

98

What means “low volume mass”? Do you   mean density?

Yes, it   was corrected.

124-127

Please reformulate. What do you mean   with “cohesive”?

Corrected.

141-148

Eliminate this paragraph

It was   deleted, what a pity. However, you are right – it is general piece of   information.  

Table 1

What means “modus”?

It was   English incorrectly used term. It should be “Mode” - https://en.wikipedia.org/wiki/Mode_(statistics)

Table 1

No digit after the comma.

It was   corrected, but I do not agree. The device enables to measure with given   accuracy, therefore digits had been stated. It was not in the review but also   units were added.

Fig. 3 and 4

What means “diameter”, since the shape   of the particles is not globular.

I   agree, it is a stupidity, it was deleted.

Fig. 5

“Diameter”: do you mean length?

Yes,   length.

Table 2

“mPa*s” instead of “MPa*s”

Thank   you, the mistake had been done when formatting.

Table 2

Use SI units

I   understand it e.g. at a density. It is information of the producer. I   would hate to intervene. The quotation was added.   

205

What means “distribution medium”?

We used   better term – Gelcoats - https://www.havel-composites.com/en/categories/epoxy-and-polyester-resins%2C-hardeners-and-gelcoats-363

214

“perfect saturation”: Do you mean good   distribution of the FCR in the MHC?

Yes,   thank you.

218-220

Please explain better. What does   “saturation” mean here?

The   text formulation was changed.  

239

What means “composite adhesive bonds of   40 wt%”? Do you mean the composite with 60% matrix and 40% filler?

Yes,   corrected.

246

“Surface”: do you mean the surface of   the steel parts which were bonded adhesively?

Yes,   added. 

239-245

Please explain better; the reviewer does   not understand what had been prepared. Add a picture or drawing of the test   samples. Where is the “lapped length”? The reader will not necessarily know   the various EN standards.

Only in 256-260 it is then explained   that you talk about shear tests.

Added,   you are right, it is not entirely known.

241

What is the thickness of the adhesive   layer? According to Fig. 20 it is approx. 100 – 110 µm. Is that correct?   Please add this information.

Yes,   you are right. But your information relates to filler FRC 20. Other fillers   showed higher thickness of the adhesive layer. Thank you for the comment.   Added. 

273

What means “hybrid” here?

Hybrid – the term used at adhesive bonds where the adhesive contains   filler, namely biological.  Similar comment was also at the reviewer 1.

279 and following lines

No digit after the comma at such high   standard deviation.

Adjusted,   but my opinion is presented above.

288-292

Less digits after the comma; so many   digits are not significant.

Corrected.

279-287

Most part of this section 2 Materials   and Methods

Corrected   according to above mentioned comments.

296, 297

“decrease” instead of “fall”

Corrected,   thank you.

293 and following

It is surprising, that mechanical   properties decrease with addition of the filler, especially also when adding   the fibrous filler. One would expect a fortification of the matrix by the   filler. Is this discussed further down in the paper?

Yes, it   is not surprising according to my opinion. It is particle filler and short   fibres. They do not usually significantly increase e.g. the strength. They   increase e.g. the hardness.  

306

How are these numbers calculated? What   is the basis for this percentage calculation?

Classical   calculation where the matric means 100.  

310

“reinforcing phase”: do you mean the   filler? “Reinforcing” is no good term, because here the results get worse   when adding the filler.

Corrected,   however, it was one of expression of the composite material definition.

309

Do you mean the surface of the filler   particles and fibers?

Filler   added – so, it is particles as well as fibres.

311

Explain this surface micro-dissolve   process.

I think   that the formulation is sufficient for this paper. There is stated the   quotation in case of interest which deals more detailed with the topic.

311

“increase of fiber strength”: Do you   mean tensile strength of the fibrous filler? Or do you mean more compact   surface of the fibrous filler, which then gives stronger bonding effect   between the fibrous filler and the matrix?

Corrected.   I agree, ambiguous formulation.

310

What do you mean with “interface” here?   Usually “interface” is just the two-dimensional contact zone between matrix   and filler, whereas the “interphase” is the joint region of matrix and filler   surface (where the molecules of the matrix might penetrate slightly into the   surface of the porous filler).

Yes,   interphase would be better according to above stated.

318

What exactly do you mean with “interface   optimization”? Please describe more in detail, because this is an important   feature of the whole work you performed.

Adjusted.   Your formulation was used. Thank you.

316

What might be the effect of these   “undesirable surface layers”? Will there be the breaking zone exactly at the   interface between matrix and filler and not a cohesive failure within the   matrix?

It is   probable that the breaking zone between the matrix and the filler occurs.  

326

You mean that the NaOH treatment   improved the wettability of the filler surface by the matrix molecules. This   sounds logical and should be mentioned in this clear mode, if you agree with   this assumption.

Thank   you, you are right again.

328

Please explain somewhat more in detail   here you exactly see this effect, that adhesive strength is higher than   cohesive strength of the FCR. Can the mentioned improvement of the fiber   strength due to the NaOH be seen somewhere on the photos in Fig. 10?

Yes, it   is evident in original fig. 10. I apologize myself, but if I wanted to describe   it better, I would need to know the filler strength and I cannot measure it.   We do not have a possibility for the filler destruction in SEM.

361

“Figure 10” instead of “Figure 9”.

Corrected.   Thank you for notice.

331

Though it seems to be obvious, please   add an arrow pointing to this hollow profile.

Indicated   in figure.

330

“reactoplastics”: please replace by any   other term; this is very strange word the reviewer never heard.

It is   thermosetting polymer  –  https://en.wikipedia.org/wiki/Thermosetting_polymer. Corrected also in next text.  

328/329

Please add some arrows to show where   exactly this destruction inside FCR layer can be seen.

Indicated.

330

Please add some arrows where exactly to   see this “brittle fracture”.

Indicated.

335

What can be an explanation that “MHC”   and “MHC-FCR700 (5% NaOH)” are equal in tensile strength at 20°C but not at   higher temperatures?

I   admit, I have no idea. However, it is an average value, but MHC-FCR700(5%   NaOH) is of higher standard deviation of results. It is evident from results   that some values of MHC are of significantly higher tensile strength at 20 °C   than the average value of MHC. 

Fig. 7 and 8

Please divide the figure into three   parts be vertical lines, with the designation of the three parts with the   relevant temperature (but not in the sense of a x-axis). It took some time to   understand that always 5 results belong to one temperature.

Adjusted,   now clearer. Thank you for notice.

338-339

Fewer digits after the comma. 2.9 and   226 % is accurate enough; more digits are not significant according to statistical   error calculation.

Corrected.

342

Why do you call it a “positive” effect   if the elongation at break increases?

Matrix   is determined for laminating and forming. Therefore the positive effect.  

Fig. 7 and 8

Please discuss possible “correlation”   between tensile strength and elongation at break, for the various materials   as well as the various temperatures. If not done later below (the reader does   not know this when reading looking at the Figures 7 and 8) calculate apparent   MOE and discuss if addition of the FCR yields fortification or rather   weakening of the system.

Thank   you. The text and figure (graph) were added. The graph was adjusted according   to the comments to fig. 7 and 8.

Figure 9

Please describe more in detail (also   adding suitable arrows in the photos) what can be seen on the pictures.

Adjusted.   I do not know what to add to the description.

366

What means “Coulomb friction” here?   Transfer of forces and stresses is possible if there is bonding between the   two materials; also static friction might help, but for this you would need   external forces to hold the two surfaces together, which is not so much the   case here.

Yes,   you are right, adjusted.  

375

Add the remark that the statistical   difference is for the change of temperature. Did you also evaluate   statistical differences at the same temperature between the various material   combinations as given in Table 3?

Added   into the description of the table. The statistical testing of materials was   not performed.

373

Looking at Figure 8 it looks like that   there is no statistical secured difference between 40 and 60°C for the two   material combinations (i) MHC-FCR100 and (ii) MHC-FCR100 (5% NaOH). Please   check your evaluation and given more details conclusion in the text.

Text is   OK, however, it was necessary to clarify it. Another statistical test was   performed, which certified your statements – p values were quantified.

378-380

What is certified, improvement or   weakening? Just indication of “affects” is not specified.

Corrected.

381-383

What does this mean?

Deleted.

384 ff.

The following tests obviously are linked   to the adhesive bond tests between two steel materials. This should be   explained more clearly, in case with a separate chapter of the paper.

It should also be mentioned shortly in   the chapter “Materials and Methods” which tests are done with which samples.   This is not clear when reading this chapter.

Text adjusted   – the material adhesion was evaluated by the fracture surface of adhesive   bonds. See comment to indication AF, SCF and CF. Cyclic tests are described   in the methodology in original lines 256 – 264. Thank you for understanding.  

405, 407

No digit after the comma.

Corrected

404

“positive way”: what are the arguments   that a higher elongation of break is    positive effect?

Deleted.

408-410

Please check if there is really   statistically significant difference between “AB-MHC” and “AB-MHC-FCR100 (5%   NaOH)” for bond strength and for elongation at break. Is it correct that p =   0.000 for all these cases?

Verified,   certified.

Table 7

The codes “AF”, “SCF”, and “CF” must be   explained. Readers will not know necessarily the details of this EN standard.  

Yes,   added, thank you for notice.  

423

“to” instead of “do”

Thank   you, corrected.

425

How can there be a viscoelastic   behaviour if the used epoxy resin is a hardening duroplastic system? Is it   viscoelastic behaviour of the adhesive layer as such or are there already   micro-cracks which cause the elongation and the creep behaviour?

The   producer states that the matrix is determined for laminating at increased   temperatures. Thank you for a topic for other tests. Microcracks were not   identified behore tests. But the test was performed only by means of SEM   images of cuts. So it is not reliable. It would be better to use   non-destructive tests. The firm Olympus has been developing a device for air   industry at the present but is not yet fully tested.  

Table 8

Number of cycles: Use only full digit   numbers for the average, no comma. It will be better to indicate the span   (lowest number – highest number of cycles) instead of the standard deviation.  

Adjusted,   the standard deviation was left. I know it would be more suitable to listen   the reviewer, but this expression seems to be more suitable for me.   

418

How is this test done? Is it a static   test (bond strength and elongation at break) similar to Figures 11 and 12,   but now performed after the 1000 cycles? Do you somewhere compare the results   before the 1000 cycles and after the 1000 cycles at the various intensity of   cycles? Figure 11 and Figure 13 could be combined, as well as Figure 12 and   Figure 14, and then you can show the influence of the pretreatment with the   1000 cycles.

Cyclic   tests are described in the methodology in original lines 256 – 264. Thank you   for understanding. I hope it is understandably. Tests are based on a static   tensile test but software of testing machine is programmed for a repetition   of tests according to the methodology on page 256 – 264. Data presented in   fig. 11 serve for setting default test parameters. However, they cannot be combined.    

433

What might be reasons for this effect?

Irregular   shape of short randomly oriented fibres.  

446, 448

It should be indicated that these are   the results of static tests after performing up to 1000 cycles as   pretreatment.

Yes, of   course. Thank you.

Figures 17 and 18

What is the difference between the two   Figures? The test conditions had been the same, but in one test the maximum   of 1000 cycles was reached without destroying the sample, and in the other   case (one of the 4 tests out of 10 as indicated in Table 8) the sample   collapsed already after 343 cycles. Is that correct? This should be mentioned   somewhere in the text for better understanding.

Yes, it   is right. The figure presents a difference between finishing 1000 cycles and   a premature destruction. It was added into the text.

Figures 15 – 17

So first the 1000 cycles were performed   and then immediately the test was performed to break (without removing the   sample from the machine between the last cycle and the test until break. Is   it like this?

Yes, it   is just you write. It is stated in the methodology. However, the adhesive   bond did not withstand 1000 cycles in some cases.  

465

What does this mean? Please discuss more   in detail.

Added.

Figures 15 – 18

What means the word “low” in “low cyclic   test”?

Tests   to 10000 cycles are indicated as low cyclic tests. Tests over this number are   indicated only as cyclic tests.

485

What do you mean with “interaction”? The   small cracks are a consequence of the dynamic tests with the 1000 cycles.

Interaction means a   contact of the filler and the matrix. It can be assumed that small cracks are   owing to dynamic tests.  

260

Why do you call it “quasi-static test”?   Is it not it rather a dynamic test? What is the duration of one cycle for the   various versions?

It is   stated like this in the software of testing machine.

483-490

This paragraph is most important; here   discussion should be more in detail.

I tried   to describe it better in accordance with above presented comments.

Table 9

What is compared in the various cases in   order to prove if statistical difference is secured? E.g. 0.0001 for adhesive   bond strength for AB-MHC: it is significant different from what?

The influence of various types of low-cyclic   fatigue, i.e. from 5 % to 70 % (208 N – 2906 N), from 5 % to 50 % (208 N – 2076   N) and 5 % to 30 % (208 N – 1246 N) was compared.

530-547

The FCR 700 (especially as 5% NaOH   version) is superior for the properties of the laminates and close to the   neat MHC. But in application as adhesive in an adhesive layer this filler was   the worst. This should be discussed more in detail.

I   presume it is caused by the loading type and by the thickness of the   composite mixture layer.

General remarks:

As mentioned several times above, too   many digits as presented after the comma in many cases. These digits are   statistically not significant

Corrected.

The whole paper needs strong improvement   of English language by a native speaker.

Handed   over to a native speaker for review.

The whole paper needs reformulation, in   order to achieve scientific style of writing. So far the text is very   complicated and tedious to read. Often text book information is given which   can be deleted. Often repetition of information is given.

I tried   to adjust it. I hope you will be satisfied.  

All mentioned EN standards must be also   part of the References.

Added.

Round  2

Reviewer 2 Report

More or less all proposed changes / coorections have been preformed by the authors.

Some small changes are still necessary (see attachment). 

Author Response

Revize 2 oponet – Müller et al Polymers 502703 Comments by Reviewer June 2019 (second review round)

Changes in the text are made by blue colour. Thank you for valuable comments. I appreciate these factual comments.

Comments

Table 1

“No digit after the comma.” I understand the opinion of the authors, that enough accuracy of measuring is as such. But with a standard deviation of approx. 60 – 70% it gives no sense to be more accurate in the measurement than 3% in the best case and 0.2% in the worst case.

Yes, I agree and I thank you for the comment. The standard deviation was removed.

Table 1

I assume that the measured numbers indicate the length (or the longer axis) of the particles; this should be mentioned clearly in Table 1 and Fig. 3 – 5, and also in the text; so far it is only roughly indicated in caption of Fig. 2.

Yes, you are right. Added into the text with the reference to the table 1 and fig. 3 – 5.

Line 207

Gelcoat: this is a (lacquering) substance applied to the surface of fiber reinforced materials. It is not clear what “gelcoat” will mean here.

Thank you for the comment. Added into the text. As an example I also add a reference https://en.wikipedia.org/wiki/Gelcoat. Gelcoats are modified resins which are applied in a liquid state on moulds and are used for securing high quality surface treatments of a composite material visible surface. 

Line 256

“The surface of the steel parts, which were bonded adhesively …”

Thank you, adjusted.

Line 287

Delete “hybrid”.

I agree. Deleted. Deleted also in other places. It was left only in one place because of cited text – line 118. 

Line 323

plniva ?

Thank you, deleted the Czech word for the filler.
